# Evolving Landscape of Paediatric Inflammatory Bowel Disease: Insights from a Decade-Long Study in North-East Slovenia on Incidence, Management, Diagnostic Delays, and Early Biologic Intervention

**DOI:** 10.3390/diagnostics14020188

**Published:** 2024-01-15

**Authors:** Martina Klemenak, Manca Zupan, Petra Riznik, Tomaz Krencnik, Jernej Dolinsek

**Affiliations:** 1Department of Gastroenterology, Hepatology and Nutrition, Pediatric Clinic, University Medical Centre Maribor, 2000 Maribor, Sloveniajernej_dolinsek@hotmail.com (J.D.); 2Faculty of Medicine, University of Maribor, 2000 Maribor, Slovenia

**Keywords:** children, ulcerative colitis (UC), Cohn’s disease (CD), management, biological therapy

## Abstract

Background: In the past decade, significant progress has been achieved in the care of children with inflammatory bowel disease (IBD). Our study concentrated on assessing the incidence and management of IBD in children in North-Eastern Slovenia over a 10-year period. Methods: Medical data from children and adolescents diagnosed with IBD in North-Eastern Slovenia (2014–2023) was analysed. Disease incidence and management of children were assessed. Findings were compared between two periods (2014–2019 and 2020–2023, coinciding with the COVID-19 pandemic). Results: 87 patients (median age 15.5 year; 50.6% male) with IBD (43.7% Crohn’s disease (CD)), diagnosed between 2014 and 2023 were included. Extraintestinal manifestations were more common in CD than ulcerative colitis (UC) (15.8% vs. 2.4%, *p* < 0.05). Median delay from symptom onset to diagnosis was 2 months, lower in UC than CD (NS). Mean annual IBD incidence per 100,000 children aged 0 to 19 years was 6.4 (95% CI 4.4–8.3), slightly lower for CD than UC (2.8/100,000 vs. 3.1/100,000). In the second period, IBD incidence significantly rose (9.1 vs. 4.6, *p* < 0.05). During this period, 53% of CD patients transitioned to biological treatment within three months of diagnosis. Conclusion: IBD incidence rose among children in North-Eastern Slovenia over the past decade. Additionally, more children with CD underwent biological therapy in the second period.

## 1. Introduction

Over the past decade, substantial progress has been achieved in the care and management of children with inflammatory bowel disease (IBD)–Crohn’s disease (CD), and ulcerative colitis (UC). This notable advancement is attributed to the high research interest in this field and expanding therapeutic options available, including more biological treatment options registered for children [1]. In Slovenia, management and treatment of paediatric IBD is guided by the recommendations of the European Society of Paediatric Gastroenterology, Hepatology and Nutrition (ESPGHAN) and European Crohn’s and Colitis Organisation (ECCO) [1,2,3].

Since 2020, the therapeutic approach to CD in children has advanced significantly, incorporating predictors of poor outcomes, and supporting the early application of anti-tumour necrosis factor therapy for patients identified as high risk of developing complicated disease [1].

In the case of UC, biological therapy is reserved for later stages in the disease course, particularly when the disease remains chronically active or exhibits 2–3 annual flares despite consistent therapy with thiopurines and aminosalicylates [2]. Therapeutic focus in children with IBD has evolved to prioritize biochemical or endoscopic remission over clinical remission, recognizing the persistence of intestinal inflammation even after the resolution of abdominal symptoms [4].

The aim of our study was to evaluate the incidence of IBD in children over a ten-year period in North-Eastern (NE) Slovenia, and delineate the phenotypic characteristics of the disease at the time of diagnosis. Additionally, we sought to investigate the evolution of the initial treatment approach over the span of a decade. This investigation was prompted by significant shifts in treatment possibilities, the emergence of new risk stratification methods for CD, and the introduction of novel biological treatments specifically designed for use in children.

## 2. Methods

This retrospective study investigated a cohort of children and adolescents under the age of 19 residing in NE Slovenia diagnosed with IBD between 2014 and 2023. We included children that were diagnosed with CD, UC, or inflammatory bowel disease unclassified (IBD-U), meeting clinical, laboratory, endoscopic, radiologic, and histopathologic criteria for IBD. Our centre conducted a complete diagnostic workup, involving upper gastrointestinal endoscopy, ileocolonoscopy, and small bowel imaging (capsule endoscopy or MR enterography) for CD and IBD-U, and upper gastrointestinal endoscopy and ileocolonoscopy for UC. Exclusion criteria comprised individuals diagnosed outside our centre. Detailed analysis of medical data was conducted, focusing on the clinical presentation, diagnostic procedures, and treatment strategies for newly diagnosed patients. 

The diagnostic process involved assessing disease activity using the Paediatric Crohn’s Disease Activity Index (PCDAI) for patients with CD and the Paediatric Ulcerative Colitis Activity Index (PUCAI) for patients with UC. Disease location, extent, and behaviour were determined using the Paris classification [5]. Only patients with a complete diagnostic workup were included in the analysis. 

Special attention was directed towards the treatment regime, with a specific emphasis on patients undergoing biological treatment and their subsequent follow-up. Two periods were compared: period 1 (2014–2019) and period 2 (2020–2023). The overall incidence of surgical interventions throughout the study period was determined. Furthermore, an assessment was made regarding the number of IBD patients necessitating biological treatment. The time interval between the confirmation of the diagnosis and the initiation of biological therapy was calculated and compared across selected time periods and between distinct disease groups.

The incidence of IBD over the past decade was calculated based on demographic data sourced from the Statistical Office of the Republic of Slovenia. In Slovenia, children with IBD are managed by two centres. Our centre, located in NE Slovenia, covers approximately one-third of the paediatric population in the country.

Statistical analysis was performed using IBM SPSS Statistics 24.0. The study was approved by the National Medical Ethics Committee of the Republic of Slovenia (0120-211/2022/3).

## 3. Results

### Patients’ Characteristics

In our study, 87 patients (median age 15.5 years; min 4 year, max 18.8 year; 50.6% male) with IBD (43.7% CD, 48.3% UC, 8.0% IBD-U), diagnosed between 2014 and 2023 were included. According to the Paris classification, the most prevalent age group was A1b (10–17 years, CD 71.1%, UC 64.3%). Family history of IBD was present in 12.6% cases. The median body mass index z-score for age at the confirmation of diagnosis was −0.45 (min −4.43, max 2.49). Patients with CD had significantly lower BMI compared to UC patients (*p* < 0.05). 

At diagnosis, the most common symptom was diarrhoea (75.9%), followed by abdominal pain (73.6%), and bloody stools (65.5%). Three patients with CD had perianal disease at diagnosis (Figure 1). A total of 18.4% of all the patients had extraintestinal manifestations (EIM) of IBD, out of which 62.5% had fever, followed by primary sclerosing cholangitis, uveitis (both 18.7%), and arthritis (12.5%). EIM were more common in patients with CD compared to UC (15.8% vs. 2.4% *p* < 0.05). Median delay from the onset of symptoms to the confirmation of diagnosis was 2 months (0–3 year). In patients with UC, median delay was slightly shorter compared to CD patients (2 m vs. 3 m, NS). Terminal ileum was reached in 97.7% of patients. Among CD patients, the disease was most commonly ileocolonic (L3-52.6%), followed by colonic disease (L2-26.3%). In 26.3% of patients, upper GI tract was also involved (L4a). In most of the patients (92.0%) the disease was non-stricturing and non-penetrating (B1). Among UC patients, the disease was most commonly presenting as pancolitis (E4–52.3%), followed by extensive colitis distally from hepatic flexure (E3–21.4%). Only 9.5% of UC patients presented with severe disease (S1). More details regarding patients are presented in Table 1.

The mean annual incidence of IBD per 100.000 children aged 0 to 19 years for the study period was 6.4 (95% CI 4.4–8.3). Incidence for CD was slightly lower compared to UC (2.8/100.000 vs. 3.1/100.000, respectively). The incidence of IBD-U was 0.5/100.000 (95% CI 0–1.1); however, this group of patients did not meet quantitative criteria for further statistical analyses.

For further analysis, we divided the observed period into two distinct timeframes: the first from 2014 to 2019 and the second from 2020 to 2023, coinciding with the onset of the COVID-19 pandemic.

During the second period, a significant rise in the incidence of IBD was observed (9.1 vs. 4.6, respectively; *p* < 0.05). Specifically, the incidence of CD rose significantly (3.8 vs. 2.1; *p* < 0.05), as did that of UC (4.2 vs. 2.6; *p* < 0.05) (Figure 2).

## 4. Treatment of IBD

### 4.1. Induction Therapy

During the first period (2014–2019), 37 patients with IBD (46.0% CD, 51.3% UC, 2.7% IBD-U) were diagnosed. Due to low number of IBD-U patients, we excluded them from further analysis.

As an induction therapy, almost half of CD patients (47%) were treated with corticosteroids, 18% with exclusive enteral nutrition (EEN) and one patient with biological therapy. Other CD patients were treated with azathioprine or mesalazine, or a combination of the two. Patients with UC were all treated with mesalazine at the induction, 63% of them were additionally also treated with corticosteroids.

During the second period (2020–2023), 50 patients with IBD were diagnosed (42.0% CD, 46% UC, 12% IBD-U). The proportion of CD patients, treated with corticosteroids at the baseline increased to 67%, and the proportion of patients treated with EEN decreased (5%) due to the introduction of a Crohn’s disease exclusion diet (CDED) and partial enteral nutrition (PEN), which became the first line therapy for 19% of CD patients. In the second period, the proportion of children treated with infliximab as a first-line therapy increased (30% vs. 7%; NS). The proportion of UC patients treated with corticosteroids was slightly lower (57%), all were given mesalazine and one patient was treated with azathioprine. No UC patient was treated with biologics at baseline.

During the first period (2014–2019), out of 37 IBD patients, only one patient with CD was treated with biological therapy (infliximab) as an induction therapy. During the second period (2020–2023), the biologicals were used as an induction therapy in seven CD and one IBD-U patient (two-times adalimumab, six-times infliximab).

### 4.2. Follow-Up Therapy

The use of azathioprine in patients with CD at follow-up did not differ between observation periods (53% vs. 62%, NS). The most notable alteration in follow-up therapy between the two periods was observed in the administration of mesalazine in CD patients, with a decrease from 41% to 0% (*p* < 0.05). Additionally, during the second period, there was a considerable reduction in the proportion of CD patients undergoing surgery compared to the first period (24% vs. 5%). Interestingly, nearly two-thirds of patients with CD were receiving partial enteral nutrition at follow-up. For patients with UC, the use of mesalazine treatment decreased; however, the proportion of azathioprine use did not differ between the two periods.

### 4.3. Biological Therapy

Altogether, 41% of IBD patients (*N* = 36) were treated with biological therapy (58% of all CD patients and 31% of all UC patients). In addition to nine patients (eight CD, one IBD-U) that were already treated with biological therapy at the induction, twenty-seven other patients underwent a transition to biological therapy during the follow-up (52% in CD and 48% in UC). A marked increase in the administration of biological treatment as maintenance therapy for CD was observed between the first (2014–2019) and the second (2020–2023) time periods (33% vs. 74%, *p* < 0.05) (Table 2).

Throughout the observed period, five patients with IBD underwent a transition to a second biological therapy and one patient underwent two therapy modifications. Among them, four individuals on infliximab developed antibodies to the drug. Additionally, one patient, initially prescribed vedolizumab for UC was switched to infliximab due to a loss of response; it is worth noting that, at that time, determination of vedolizumab levels was not available in our country.

In the subgroup of nineteen patients with CD receiving infliximab, only one patient underwent a switch to a second biological treatment, specifically ustekinumab. In contrast, among the ten patients with UC receiving infliximab, three experienced a shift to a second biological treatment—two to vedolizumab and one to adalimumab (Table 3).

### 4.4. Initiation of Biological Therapy

The median time interval from the confirmation of the diagnosis of IBD to the initiation of biological therapy during the study period was 7 months (IQR 2.5–19.3 months; 5 months for CD, 12 months for UC, NS). Notably, a significant difference in the time from diagnosis to the commencement of biological treatment was observed between the two study periods, with the interval significantly shorter during the second period (6 months (IQR 2–16) vs. 30.5 months (IQR 6.3–35.8); *p* < 0.05). Specifically, during the second period, 53% of CD patients transitioned to biological treatment within three months from the diagnosis. This percentage was significantly higher than that observed in the first period (53% vs. 20%; *p* < 0.05).

## 5. Discussion

Our retrospective study outlines the incidence, phenotypic characteristics, management, and treatment of inflammatory bowel disease in children in North-Eastern Slovenia during the ten-year period from 2014 to 2023.

### 5.1. Incidence of IBD

Notable increase in the incidence of IBD in children was observed over a 10-year period, aligning with global trends observed in the epidemiology of IBD [6,7,8,9,10,11,12,13]. The mean annual incidence of IBD in North-East Slovenia from 2014 to 2023 was slightly lower compared to the study conducted by Urlep et al. [10] in the same region from 2002 to 2010 (6.4/100.00 vs. 7.6/100.000) and higher compared to Orel et al. [9] for the study period from 1994 to 2005 focusing on Central and Western Slovenia (4.03/100.000). Additionally, our study identified a significant surge in incidence between 2020 and 2023 (9.1/100.000), as opposed to the period from 2014 to 2019 (4.6/100.000). This increase coincided with the onset of the COVID-19 pandemic. Similar trends were observed in the study from Ashton et al. [14] from the south of England and Rosenbaum et al. [15] from New York City (NYC), both reporting a rise in incidence at the beginning of the 2020 pandemic. These findings prompt inquiries into the potential association between viral illnesses, particularly SARS-CoV-2, and the pathogenesis of IBD. However, further research is crucial to comprehensively understand these results.

A comprehensive review by Sykora et al. [8] revealed that CD tends to predominate over UC and IBD-U in regions with a high incidence of IBD. However, exceptions exist in areas like Eastern Europe, where the incidence of UC surpasses that of CD. Inconsistencies in data regarding the CD and UC ratio were observed in our study and two other previous studies in Slovenia. The mean annual incidence of CD in our study was higher compared to the period of 1994–2005 (2.8/100.000 vs. 2.42/100.000) and lower compared to 2002–2010 (2.8/100.000 vs. 4.6/100.000). Regarding UC, our study found a higher mean annual incidence compared to the periods 1994–2005 and 2002–2010 (3.1/100.000 vs. 1.14/100.000 vs. 2.8/100.000) [9,10]. This variance raises questions about the changing dynamics of CD and UC incidence over time and underscores the importance of ongoing research to understand these patterns in the context of regional and temporal factors.

### 5.2. Age at Diagnosis of IBD

In our study, the median age at diagnosis was 14.8 years for CD and 15.9 years for UC, which is higher compared to other studies [16,17,18]. Notably, there were no significant differences observed in median age or distribution within the age groups according to the Paris classification. It is recognised that the incidence of IBD in children is highest during adolescence, which is consistent with our data. Within the age group below 10 years (A1a according to the Paris classification), the incidence is around 18% [19]. However, in our study, this group represented 7.9% and 7.1% for CD and UC, respectively, which is lower than reported in other studies [13,19,20]. These variations underscore the importance of considering demographic and regional factors that may influence the age distribution of paediatric IBD and highlight the need for further research to better understand these differences.

### 5.3. Clinical Presentation of IBD

The most common symptoms at the time of IBD diagnosis were diarrhoea (75.9%), abdominal pain (73.6%), and bloody stools (65.5%), which is similar to recent findings of Pivac at al. in Croatia [16]. Both studies also found that bloody stools were significantly more common in UC. It is known that CD can lead to linear growth retardation, whereas growth impairment is less frequently associated with UC [21]. Consistent with this, we found that weight loss and growth retardation were more common in CD, although the results weren’t statistically significant. Patients with CD had a significantly lower BMI compared to those with UC. While other studies have reported a higher incidence of growth impairment in CD patients, the results have been inconsistent regarding statistical significance [16,22].

The prevalence of EIM in children with IBD at the time of diagnosis has been reported to be up to 28% [23], with some studies indicating a higher overall prevalence over the course of the disease [24,25]. Our study revealed that 18.4% of all IBD patients had at least one EIM, a finding comparable to the data reported by Greuter et al. [26], where the occurrence of EIM was 16.7%. EIMs were significantly more common in CD compared to UC in both our study and the study of Greuter et al. [26] (15.8% vs. 22.5% for CD and 2.4% vs. 10.3% for UC, respectively), a pattern also observed in other studies [23,27,28]. However, contrary results have been reported in the study by Adam et al. [29], where EIMs were more common in UC than CD, and in the study by Jose et al. [30], which found no correlation with the subtype of IBD. These divergent findings underscore the complexity of EIMs in paediatric IBD and highlight the need for further research to elucidate the factors influencing their occurrence and subtype-specific prevalence.

### 5.4. Diagnostic Delays in Paediatric IBD

The median diagnostic delay in our study was found to be 3 months for CD and 2 months for UC, consistent with findings from other studies that commonly report longer delays for CD compared to UC [17,18,31,32,33,34,35,36,37]. Recent investigations from El Mouzan et al. [31] and Sulkanen et al. [17] found median delay for CD and UC at 8 and 5 months in Saudi Arabia, and 6.6 and 4.1 months in Finland, which is slightly longer compared to our study. Additionally, a systematic review by Ajbar et al. [32] revealed an overall median delay range of 4–24 months for CD and 2–18 months for UC. One potential explanation for the earlier diagnosis of UC is the higher occurrence of bloody stools, which often raises more immediate concern compared to the nonspecific symptoms associated with CD, such as weight loss and fever [17,37]. This might contribute to earlier referral of patients with UC to gastroenterologists. A study by Ricciuto et al. [36] performed in Canada identified a “referral delay” as the primary contributor to the overall diagnostic delay, emphasizing the importance of timely referrals in expediting the diagnostic process.

### 5.5. Induction Therapy

The updated Crohn’s Disease treatment guidelines in 2020 prompted a modification in the induction treatment protocol at our institution. The guidelines recommend nutritional therapy for induction therapy in cases of purely inflammatory disease behaviour and low-to-medium risk at the time of diagnosis [1]. This change has resulted in a noticeable shift in the patient population receiving exclusive enteral nutrition (EEN), particularly with the integration of the Crohn’s Disease Exclusion Diet (CDED). The CDED involves a whole-food diet combined with partial enteral nutrition (PEN) and is designed to minimize exposure to dietary components that may have adverse effects on the microbiome and intestinal barrier [38]. This approach is supported by compelling studies demonstrating comparable efficacy [38,39]. Post-2020, the introduction of CDED as a therapy shown to be as effective as EEN has led to a reduction in the use of EEN in the second study period (18% vs. 5%), with CDED being more readily tolerated and replacing EEN (19%). This shift reflects an evolving understanding of dietary interventions in the management of CD and a move towards treatment strategies that are not only effective but also more palatable for patients.

Treatment with corticosteroids is specifically reserved for children with active luminal Crohn’s disease when exclusive enteral nutrition is not a viable option, or for low to median risk patients [1]. Interestingly, in the second period, there was a notable increase in the percentage of CD patients treated with corticosteroids (from 47% to 71%), despite the introduction of diet therapy and the swift adoption of biologics. This increase is likely attributed to the established orientation of our centre, built on prior positive experiences with corticosteroid therapy for remission induction, a trend also observed in other medical centres [40,41,42]. This underscores the influence of historical practices and institutional preferences in shaping treatment approaches, even in the presence of evolving guidelines and the introduction of novel therapies.

The recent ESPGHAN and ECCO guidelines recommend initiating biological treatment from the outset if there is growth delay, for individuals at high risk of poor outcomes, and after treatment failure with EEN or corticosteroids, to achieve remission [1]. In alignment with these guidelines, there was a statistically significant increase in the utilisation of biologics for the induction therapy of CD during the second period (2020–2023), rising from 6% to 38% compared to the preceding period (2014–2020). This shift reflects the evolving treatment landscape and the prioritisation of biologics as an effective and targeted approach in instances where growth delay is evident or when conventional therapies prove insufficient.

In the management of UC in children, the treatment guidelines have not undergone significant changes in recent years. Typically, biological treatment should be considered in chronically active or steroid-dependent UC, uncontrolled by aminosalicylates [2]. Nevertheless, newer studies have shown that biological therapy is having an important role in the treatment of acute severe colitis [4]. Our study showed that almost one-third of patients with UC received biological treatment during the course of the disease, which is slightly higher than in the study by Kaplan et al. [43], where 25% of patients with UC were switched to biologics.

### 5.6. Follow up Therapy

The most significant change in follow-up therapy for CD during the second period was the discontinuation of treatment with mesalazine, with a decrease from 41% to 0%. This change is supported by the lack of evidence to endorse the use of mesalazine for maintenance therapy in children with CD [44,45]. However, in the second period, a subset of patients (14%) with mild colonic involvement still received mesalazine as adjuvant therapy for to achieve remission. In a recent study by Abu Hana at al. [46], thiopurines demonstrated both safety and efficacy, with 21% of children with CD and 27% of those with UC exhibiting positive outcomes. These findings support the continued consideration of thiopurines as a viable treatment option for selected children with mild-to-moderate inflammatory bowel disease, particularly in cases without identifiable risk factors for a complicated disease course. This underscores the importance of tailoring treatment approaches based on individual patient characteristics and the evolving evidence supporting different therapeutic options. Indeed, thiopurines continue to be a viable option in the treatment algorithm for mild-to-moderate IBD, particularly in girls where the risk for lymphoma associated with thiopurine use is lower [47].

Biologic therapies have proven effective in inducing and maintaining remission in paediatric patients with IBD. In our study, 41% of IBD patients (58% CD, 27% UC) received treatment with biologics, a percentage similar to the study by Kaplan et al. [43] where 43% of IBD patients (50% CD, 25% UC) were treated with biologics.

However, some children may not respond adequately or may lose response over time, necessitating a switch to another biologic treatment. In our study, 5.5% of IBD patients (two individuals) were primary non-responders to infliximab, and 2.7% to vedolizumab. These findings align with the study by Kaplan et al. [43], who analysed data from the ImproveCareNow Network (*N* = 7585 children on biological treatment) and reported similar rates of primary non-response to infliximab. Additionally, 5.5% of patients in our study lost response after two years of therapy. At our centre, anti-tumour necrosis factor (anti-TNF) therapies are the first-line treatment for both CD and UC. UC patients, however, more frequently required a second biologic due to the development of antibodies compared to CD patients. While it is hypothesised that the use of biologics may reduce the number of UC patients requiring colectomy, recent published studies have yielded mixed results [48]. In our study, we had no patient requiring colectomy among UC patients. The study by Lipskar [49] reported that 9% of UC patients would require surgery during childhood. However, in our patient cohort, none underwent surgical treatment.

## 6. Conclusions

Our comprehensive decade-long data analysis indicates a rising incidence of IBD among children living in North-Eastern Slovenia. Notably, we observed relatively short diagnostic delays, a positive finding given the significant impact that prolonged delays can have on disease progression and long-term outcomes. Timely diagnosis is crucial, providing a critical window to initiate disease-modifying therapies and prevent irreversible bowel damage [50].

Despite achieving prompt diagnosis and evident treatment success, including low rates of switching biologic therapy types and surgical interventions, our study has limitations due to its retrospective nature. Our centre encompasses roughly one-third of the paediatric population in the country, raising the possibility that certain patients with more severe conditions might have sought diagnosis and treatment at the other IBD centre in Slovenia.

To address this limitation, further prospective studies are needed. These studies should focus on diverse patient variables to enhance our understanding and facilitate more efficient and targeted management strategies for children with IBD. Prospective research will provide a more comprehensive perspective on the disease characteristics, contributing to improved care and outcomes for this population.

## Figures and Tables

**Figure 1 diagnostics-14-00188-f001:**
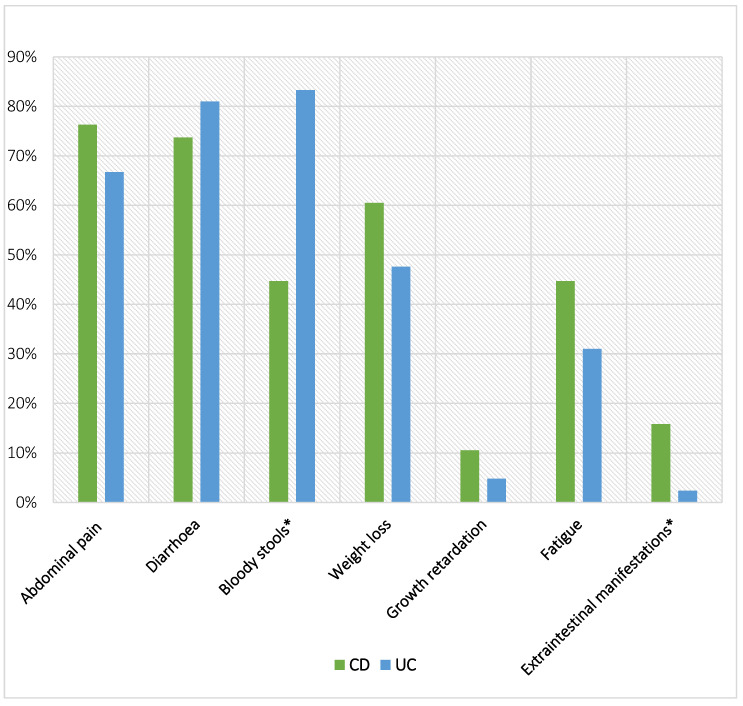
Clinical presentation of children with IBD in NE Slovenia (* *p* < 0.05).

**Figure 2 diagnostics-14-00188-f002:**
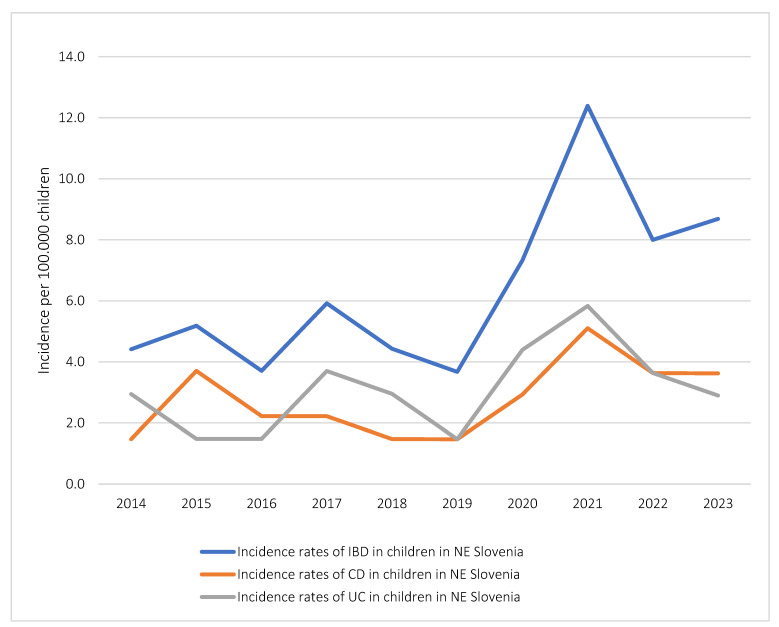
Incidence rates of IBD, CD, UC in children in NE Slovenia.

**Table 1 diagnostics-14-00188-t001:** Characteristics of included IBD patients.

	Crohn’s Disease (*N* = 38)	ULCERATIVE Colitis (*N* = 42)	Sig.
Age at diagnosis (median; min-max)	14.8 year (8 year–18.8 year)	15.9 year (4 year–18.3 year)	NS
Sex (% male)	57.9%	38.1%	NS
Paris classification (%)			
A1a (0–10 year)	7.9%	7.1%	NS
A1b (10–17 year)	71.1%	64,3%	NS
A2 (>17 year)	21.1%	28.6%	NS
Positive family history of IBD	21.1%	7.1%	NS
Diagnostic delay (median; min-max)	3 m (0–3 year)	2 m (0–2 year)	NS
Diagnostic workup (%)			
Terminal ileocolonoscopy	94.7%	100%	n/a
Esophagogastroduodenoscopy	100%	83.3%	n/a
MRE/capsule endoscopy	60.5%	21.4%	n/a
Laboratory findings			
Increased SR	54.2%	41.7%	NS
Anaemia	40.5%	49.5%	NS
Hypoalbuminaemia	11.1%	4.2%	NS
ASCA	57.9%	7.1%	*p* < 0.05
pANCA	18.4%	54.8%	*p* < 0.05
Calprotectin level (µg/g)			
<250	18.4%	21.4%	NS
250–500	23.7%	23.8%	NS
>500	44.7%	47.6%	NS
Data not available	13.2%	7.1%	n/a
Disease activity index (median)	30	35	n/a

Incidence of IBD.

**Table 2 diagnostics-14-00188-t002:** Changes in therapy between the two observation periods.

	CD (*N* = 38)			UC (*N* = 42)			IBD-U (*N* = 8)
	Period 1 (*N* = 17) (2014–2019)	Period 2 (*N* = 21) (2020–2023)	Sig.	Period 1 (*N* = 19) (2014–2019)	Period 2 (*N* = 23) (2020–2023)	Sig.	Period 1 + 2 (2014–2023)
At diagnosis							
EEN	18%	5%	NS	0%	0%	n/a	0%
CDED + PEN	n/a	19%	n/a	0%	0%	n/a	29%
Corticosteroids	47%	67%	NS	63%	57%	NS	29%
Biologics	7%	30%	NS	0%	0%	n/a	14%
Mesalazine	29%	14%	NS	100%	100%	n/a	86%
Azathioprine	41%	24%	NS	0%	4%	NS	14%
Follow-up							
PEN	no data	71%	n/a	n/a	n/a	n/a	29%
Azathioprine	53%	62%	NS	29%	27%	NS	0%
Mesalazine	41%	0%	*p* < 0.05	76%	91%	NS	29%
Biologics	33%	74%	*p* < 0.05	35%	32%	NS	14%
Surgery	24%	5%	NS	0%	0%	n/a	0%

**Table 3 diagnostics-14-00188-t003:** Characteristics of IBD patients requiring a biological therapy switch.

Patient No.	1	2	3	4	5	6
Disease type	CD	UC	UC	UC	UC	UC
Age at diagnosis	10 year 6 m	13 year 10 m	14 year 0 m	16 year 7 m	17 year 7 m	9 year 11 m
Diagnostic delay (from first symptoms to diagnosis)	6 m	2 m	5 m	2 m	6 m	1 m
PCDAI/PUCAI index at diagnosis	45	25	15	40	55	50
Time from diagnosis until biologics (months)	35 m	17 m	12 m	6 m	3 m	27 m
Type of biologic	IFX	IFX	IFX	VDZ	IFX	IFX
Time to second biologic (months)	22 m	26 m	6 m	6 m	6 m	8 m (5 m to third biologic)
Indication for the change	Ab	Ab	Ab	primary nonresponse	Ab	nonresponse
Change of biologics	IFX → UST	IFX → ADA	IFX → VDZ	VDZ → IFX	IFX → VDZ	IFX→ ADA →UST

## Data Availability

The data presented in this study are available on request from the corresponding author.

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
