# Peer review of "Evolving Landscape of Paediatric Inflammatory Bowel Disease: Insights from a Decade-Long Study in North-East Slovenia on Incidence, Management, Diagnostic Delays, and Early Biologic Intervention"

_diagnostics, 2024, doi:10.3390/diagnostics14020188_

Round 1

Reviewer 1 Report

Comments and Suggestions for Authors

This is a comprehensive study looking at different aspect of newly diagnosed patients with IBD. 

There are many similar articles in the literature. I would recommend to the authors to focus on one that is novel.

For instance difference in pre and postCOVID period or looking whether ESPGHAN guidelines are followed etc. 

Author Response

Dear reviewer,

Thank you for reviewing our article. After a literature review, we could not find many similar articles on this topic. In the article we pointed out that there were significant changes in incidence in the pre- and post-covid period and how revised ESPGHAN guidelines affected our treatment.

Our study identified a significant rise in incidence between 2020 and 2023 (9.1/100.000), as opposed to the period from 2014 to 2019 (4.6/100.000). This increase coincided with the onset of the COVID-19 pandemic.

In the second period (2020-2023), when the ESPGHAN guidelines changed, there was a notable increase in the percentage of CD patients treated with corticosteroids (from 47% to 71%), despite the introduction of diet therapy and the swift adoption of biologics. In alignment with ESPGHAN guidelines, there was observed statistically significant increase in the utilization of biologics for the induction therapy of CD during the second period (2020-2023), rising from 6% to 38% compared to the preceding period (2014-2020).

With kind regards,

Martina Klemenak

Reviewer 2 Report

Comments and Suggestions for Authors

Dear Authors, 

Your manuscript covers an interesting topic and data are clearly and very well presented. Here are just a few comments:

2. Methods 

- first sentence should be rephrased

- please state here according to which criteria the diagnosis were made and what the complete diagnostic workup include

- explain why the number of patients diagnosed in your centre is representative for the region

3. Results 

Patients’ characteristics 

-        If the sentence starts with a number, it should be written in letters not in digits. Or rephrase. 

4. Discussion 

Induction therapy 

-        biological treatment should be considered in chronically active or steroid-dependent UC, uncontrolled by 5-ASA and not only in patients who do not respond to mesalazine alone or in combination with azathioprine 

Author Response

Dear reviewer,

Thank you for taking the time to review and comment on our article.

In methods:

- first sentence was changed

- added detailed inclusion and exclusion criteria and what was included in the complete diagnostic workup

- In Slovenia, the management of children with Inflammatory Bowel Disease (IBD) are managed by two centres. Our centre covers approximately one-third of the paediatric population in the country.

In Results:

- first sentence rewritten

In Discussion:

- revised sentence as proposed on the use of biologics in UC

With kind regards,

Martina Klemenak

Reviewer 3 Report

Comments and Suggestions for Authors

Retrospective nature of study is weakest link in study.

Inclusion/exclusion criteria needs to more systemic.

Radiological role if any

Author Response

Dear reviewer,

Thank you for taking the time to review and comment on our article.

In the methods section, we added detailed inclusion and exclusion criteria, specifying the components of the complete diagnostic workup. The role of radiology of diagnosing IBD was limited to assessment of inflammation of small bowel using MR enterography.

With kind regards,

Martina Klemenak